# Vitamin A Fortification Quality Is High for Packaged and Branded Edible Oil but Low for Oil Sold in Unbranded, Loose Form: Findings from a Market Assessment in Bangladesh

**DOI:** 10.3390/nu13030794

**Published:** 2021-02-28

**Authors:** Svenja M. Jungjohann, Gulshan Ara, Catia Pedro, Valerie M. Friesen, Mansura Khanam, Tahmeed Ahmed, Lynnette M. Neufeld, Mduduzi N. N. Mbuya

**Affiliations:** 1Global Alliance for Improved Nutrition, 1202 Geneva, Switzerland; cpedro@gainhealth.org (C.P.); vfriesen@gainhealth.org (V.M.F.); lneufeld@gainhealth.org (L.M.N.); mmbuya@gainhealth.org (M.N.N.M.); 2International Centre for Diarrhoeal Disease Research, Dhaka 1212, Bangladesh; gulshan.ara@icddrb.org (G.A.); mansura.khanam@icddrb.org (M.K.); tahmeed@icddrb.org (T.A.)

**Keywords:** food fortification, edible oil fortification, vitamin A, market assessment, fortification quality, fortification compliance, Bangladesh

## Abstract

Although mandatory fortification of oil with vitamin A is efficacious, its effectiveness can be compromised by suboptimal compliance to standards. In this study, we assessed (1) the availability of oil brands across the eight divisions of Bangladesh, (2) fortification quality (the extent to which vitamin A content is aligned with fortification standards) of oil brands and producers and (3) the market volume represented by available edible oil types. We visited different retail outlets in rural and urban market hubs to ascertain available oil brands and bulk oil types and collected samples. We used high-performance liquid chromatography to quantify average vitamin A content and compared them to the national oil fortification standards. Among the 66 packaged brands analyzed, 26 (39%) were not fortified, and 40 (61%) were fortified, with 28 (42%) fortified above the standard vitamin A minimum. Among the 41 bulk oil type composites analyzed, 24 (59%) were not fortified, and 17 (41%) were fortified, with 14 (34%) fortified below and 3 (7%) fortified above the standard minimum. Vitamin A fortification is high for packaged and branded edible oil but low for oil sold in unbranded, loose form. As bulk oil makes up a large proportion of the oil market volume, this means the majority of the oil volume available to the population is either not (25%) or fortified below the standard requirement (39%). Regulatory inspections of producers selling bulk oil should be prioritized to support and incentivize the industry to make all oil traceable and fortified to standard.

## 1. Introduction

While the fortification of staple foods with micronutrients has been shown to improve nutritional and functional outcomes and be cost-effective [1,2], it is often implemented and delivered suboptimally, thereby limiting its potential for impact [3]. In contexts where fortification is mandatory, industries that process staple foods are required by law to add fortifiers to the food vehicle. The desired increase in the micronutrient intake of the target population can only be achieved through fortification if the food vehicle is consumed in a fortifiable (i.e., industrially processed) form, transported and stored under conditions that support retention of the added micronutrients in the food vehicle, and if micronutrients were added in appropriate quantity and form in the first place [2]. Even when there is very high use of a fortifiable food vehicle, the potential to benefit is often compromised by low compliance with fortification requirements [3].

In many countries, the regulatory and program monitoring capacity needed to identify and address these bottlenecks is weak due to low prioritization of the activity, lack of standardized tools and limited human and financial resources to execute [4], resulting in monitoring data not being routinely generated, accessed and used to ensure compliance [3,4]. As such, suboptimal program performance is not identified and addressed in a timely fashion. Data from different points of the supply chain can guide corrective decisions by program implementers and policymakers. Ideally, this is ensured through continuous monitoring of the fortification process by producers, as well as through inspections by regulatory authorities at the production, import and/or market sites (i.e., regulatory monitoring).

One-way of filling this data gap is through a market assessment of the availability and fortification quality (the extent to which nutrient content is aligned with fortification standards) of mandated foods at the market level. The market fortification assessment methodology is a key part of the fortification assessment coverage toolkit (FACT) [5,6] that provides standardized methods for collecting, analyzing, and presenting program-relevant data on fortification quality, coverage, and consumption of food vehicles at the market and household levels. Market assessments have been conducted as a standalone data collection endeavor in Mozambique [7], Burkina Faso [8], and Pakistan [9], Mexico [10] and in conjunction with household coverage and consumption surveys in Pakistan [11], Nigeria [12,13] and Afghanistan [14].

In Bangladesh, national fortification with vitamin A of all industrially produced edible oils, regardless of sales modality or packaging, supplied to the population was made mandatory in 2013 [15] as one of the strategies to address vitamin A deficiency [16]. This choice was based on data that suggests high coverage of the population with fortifiable oil, i.e., oil that is industrially processed and for which fortification is consequently feasible and cost-effective [17,18,19]. To ensure that the oil, in fact, contains and delivers the desired amount of vitamin A to the consumer, regulatory monitoring is needed to identify and address any issues in the supply chain. However, the regulatory monitoring capacity of edible oil in Bangladesh is limited by the multiple responsibilities of the regulatory authority and the unavailability of resources to perform regular inspections. Consequently, there are limited data on the proportion of the oil market volume and brands that are fortified according to fortification standards (i.e., fortification quality).

We conducted a market fortification assessment in 2017 across the eight divisions of Bangladesh to fill the data gap and understand more completely the quality of fortification as represented by market level indicators. Towards this end, the objectives of the study were to: (a) assess the availability of packaged edible oil brands and their respective producers as well as bulk oil types in urban and rural markets, (b) assess the vitamin A content of available edible oil brands and bulk oil types and their fortification quality compared to the national oil fortification standard, and (c) estimate the market volume represented by each of the available edible oil brands and bulk oil types.

## 2. Materials and Methods

### 2.1. Design and Sampling

We conducted a cross-sectional market assessment in Bangladesh over the period March to June 2017. We used a three-stage sampling design to select retail outlets in urban and rural market hubs across all eight divisions of Bangladesh (Barisal, Chittagong, Dhaka, Khulna, Mymensingh, Rajshahi, Rangpur and Sylhet). In the first stage of sampling, we defined and identified market hubs as centers of trading in a region of space that usually serve as main pass-through locations. These represent nodes of the main supply routes for the food vehicles, where the food vehicle is sold or passed through and dispatched to other places. Typically, volumes of goods traded in market hub systems far exceed goods traded in other systems and are characterized by the high availability of goods in one place. We purposively selected one urban city, one urban town, and three rural village market hubs in each division based on population size and density, geography and road networks.

In the second sampling stage, we listed marketplaces in each of the selected market hubs. Marketplaces are agglomerations of all types of vendors or retail outlets in a relatively large, contiguous geographic area within a market hub that allows buyers and sellers of the food vehicle to interact. In the selected rural market hubs, the main marketplace was selected. Retail outlets of the same type but located in different marketplaces may have a different product range due to differences in customer characteristics and their demand as well as the product range of their wholesaler/suppliers. To capture this diversity, marketplaces located in differing areas in a market hub were visited. The urban market hubs were subdivided into three different zones defined by perceived/apparent heterogeneity in socioeconomic status (high socioeconomic status residential areas, areas that were defined as a slum or low socioeconomic status and the third areas being other than high or low socioeconomic status). In urban market hubs, three marketplaces were selected based on the following criteria: size, the estimated number of people the marketplace serves, geographical location, and socioeconomic status of the majority of the population living in the area. To facilitate these decisions, key informant interviews were conducted to gather information on the importance and socioeconomic status of the marketplaces and available mapping data were collected [20,21].

In the last stage of sampling, retail outlets within each of the selected marketplaces were selected to be visited. The retail outlet is a general term used here to refer to different vendors or businesses that sell edible oil. Each retail outlet type can stock a different product range due to the different suppliers they use and the clientele they serve. Three different retail outlet types of the oil supply chain were defined as (1) retail shop: a small-scale shop offering a variety of goods to a local community, e.g., a convenience store or stall located in the street or are concentrated in a bazaar; (2) supermarket: a large store that sells a wide variety of goods placed in specific departments; and (3) wholesaler/distributor: an intermediary entity in the distribution channel that buys goods in bulk and sells to resellers rather than to consumers. Retail outlets of each of the three different retail outlet types were purposefully selected in each marketplace at the discretion of the data collection team and with the aim to maximize the coverage and diversity of retail outlets within each type and thereby capture the maximum of the oil brand offer. The selection and visits of retail outlets continued until no new brands were identified in two consecutive retail outlets. For supermarkets, at least one branch of each supermarket chain in the marketplace was visited.

### 2.2. Data Collection

Two teams of trained enumerators conducted the data and food sample collection under the supervision of a team supervisor. Data were collected on paper forms (retail outlet, brand, and oil specimen registration forms), which were pilot tested before the data collection started.

In each retail outlet visited, all available edible oil brands and oil types sold in bulk were listed on a brand registration form, and the following information was recorded when provided: brand name or bulk, oil type, production sites and/or supplier (distributor, exporter, importer, packer), packaging types and sizes, price, and the existence of a fortification label and/or statement.

A target of 12 samples from different batches of production or retail outlets were collected for each registered oil brand across divisions where it was available or for a bulk oil type by division [22]. Sealed consumer packs containing a minimum of 300 mL were purchased as samples of packaged oil, and samples of 300 mL or more were purchased from open oil containers of each oil type. All samples were kept in the original packaging, except for the oil sold by weight or volume. Before taking the oil samples from an oil container, the oil was mixed to ensure homogeneity and then transferred into a clean plastic container with a tight cap to prevent leaking and cross-contamination. Each sample was labeled with a unique sample code and recorded on the sample register with sample code, brand name, oil type, production sites and/or supplier (distributor, exporter, importer, packer), packaging type and size, price, the existence of a fortification label and/or statement, the batch number, production and best before date. The samples were kept in black plastic bags that were placed in cardboard boxes to ensure protection from sunlight and heat, avoid contamination and maintain good storage conditions until they were shipped to the laboratory for analysis.

Ethical approval to conduct the survey was obtained from the institutional review board (IRB) of the International Centre for Diarrheal Diseases and Research, Bangladesh (icddr,b). Retail outlet staff were informed of the nature of the assessment, and verbal consent was obtained prior to data and food sample collection.

### 2.3. Food Sample Analyses

Composite samples were prepared by brand and bulk oil type from the single oil samples and analyzed for vitamin A content at the icddr,b laboratory in Dhaka, Bangladesh. Equal parts of each individual sample of the same brand or for bulk oil of the same oil type, geographic division and source were mixed to form brand or bulk oil type, division and source-specific, composite samples. High-performance liquid chromatographic (HPLC) was used to determine the vitamin A (retinol) content in the composite samples (AOAC Method No 960.45, AOAC 2001.13 Annexure-V). Average content for each brand or bulk oil composite as compared to the national oil fortification standard (15–30 Retinol equivalents (RE) ppm in oil) [23,24,25,26]. The comparison of the measured vitamin A content with the fortification standard was done, assuming the standard range defines the desired true average vitamin A content. The oil meets the vitamin A standard if the lab result considering its measurement uncertainty overlaps with the true standard range. Each oil brand or bulk oil composite was thus classified into one of 3 fortification quality categories; (1) not fortified, (2) fortified below minimum standard (<15 RE ppm) [23,24,25,26], or (3) fortified above the minimum standard (≥15 RE ppm).

Lab results are estimates of the true content of the measurand (vitamin A) in a sample and are subject to measurement uncertainty (MU), which was determined as follows: Sum of the bias (systematic error) and imprecision (random error), both, of which can be expressed as standard deviations (SD) or the coefficient of variation (CV) = SD/mean value of the reference quantity value or the mean lab measurement, respectively [27]. The imprecision of the estimates can be expressed based on the principle of the Gaussian distribution, where 95% of the results fall within ±1.96 * SD of the mean value or ±1.96 * CV.

Reference samples spiked with two known concentrations of vitamin A (5.1 RE ppm (17,000 IU/kg) and 9.9 RE ppm (33,000 IU/kg)) were prepared by an independent laboratory and provided to the lab conducting the analysis. During the analysis of all samples, the lab analyzed one of 12 spiked reference samples at regular intervals and reported an average vitamin A content of 1.88 RE ppm and 3.30 RE ppm for the two reference samples, respectively. This corresponds to an average bias of 65% (35% recovery) between the lab result and the true vitamin A content, while the CV was 4%. Due to the large bias, we reanalyzed the quality-controlled samples independently and confirmed the spiked content of the two reference samples. Since the lab showed a low variation but systematically underestimated the quality controls samples, we applied a correction factor of 2.86 to all lab results to adjust for the bias. After correction, with 95% confidence, the true vitamin A content falls within ±8% of the adjusted lab result.

Vitamin A degradation in oil is associated with lipid peroxidation. Peroxide value was analyzed for the main oil brands and types of bulk oil to determine the degree of oxidation that may be associated with low vitamin A content. The peroxide value was measured in oil using a titration method (AOCS, 1998; method No. Cd 8-3, AOAC 965.33). The peroxide value analysis was performed in triplicate for each composite sample analyzed, and the average peroxide value was calculated. An average CV of 8% and a MU with a 95% confidence level of ±16% was determined and considered when comparing to the oil standard maximum peroxide value.

### 2.4. Data Analysis

Data were analyzed using SPSS (v 20.0, Chicago, 2005) and MS Excel (Office 2016). Summary statistics were calculated to determine the average vitamin A content and fortification quality category by brand or bulk oil composite. To present results by market volume, the results were weighted by the estimated annual oil supply volume. Annual production volumes were estimated based on a triangulation of the available oil market data [28], individual producer’s production capacity and capacity utilization or the latest annual production volume reported by them. Brand volume shares were derived from the estimated proportion of the producer’s total annual production.

## 3. Results

### 3.1. Availability, Types and Sources of Edible Oil in Bangladeshi Markets

A total of 97 packaged edible oil brands and 3 bulk oil types were available in one or several of the 553 retail outlets (50 wholesalers, 34 supermarket branches, 469 retail shops) that were visited across the eight divisions of Bangladesh.

Seventy-three of the packaged brands (75%) were locally produced, and most of them were soybean (38), 19 rice bran and 17 palm oil brands. Twenty-four brands were imported from various countries (including Malaysia, Turkey, Italy, India, Canada, and Spain), 18 of the imported brands were sunflower oil. Most of the local brands labeled with producer or supplier information were from Dhaka (38 brands), followed by Sylhet and Rajshahi (9 brands each), Chittagong (7 brands), Khulna (5 brands), Mymensingh (4 brands) and Rangpur (1 brand) These local brands were primarily sold through small retail shops (95%), rather than supermarkets (only 34% of the local brands), while the available variety of all imported brands were similar in retail shops and supermarkets (81% and 76%, respectively). Dhaka division had the largest variety of available brands (46 brands), followed by Chittagong (44 brands), Sylhet (43 brands) and Rajshahi (36 brands). Twenty-two different brands were available in Barisal, 21 in Mymensingh, 20 in Rangpur and 19 brands in the Khulna division (Figure 1). In Dhaka, Mymensingh, Chittagong and Rajshahi, the number of imported brands were higher, which reflects in the higher number of sunflower oil brands available in these divisions, which are all imported (Figure 1). The variety of available brands was higher in more populated urban areas (cities and towns) compared to rural areas (villages). Sixty-one brands of all packaged oil brands (63%) were only available in urban areas, and 8 brands were only available in rural markets (8%).

Of the 97 packaged oil brands, 52 brands were available in only one division and 45, 33, 26, 18, 13, 9, 8 brands were present in more than 1, 2, 3, 4, 5, 6, and 7 divisions, respectively. The eight brands available across divisions were produced by 6 large oil groups, while most of the smaller local brands (75%) were available in 3 or fewer divisions. The eight main brands that are available across all divisions were also found in both rural and urban areas.

The three types of bulk oil were soybean, palm oil, and super palm oil and were available across all divisions and in urban as well as in rural areas, except super palm oil that was not present in Rangpur. The minimum price of bulk palm oil, soybean, and super palm oil was 10, 18, and 20% cheaper than the corresponding type of branded oil, respectively.

### 3.2. Fortification Quality

Of the 66 packaged brands that were analyzed for vitamin A content, 39% were not fortified, 42% were fortified above the standard minimum (≥15 RE ppm) [23,24,25,26] with an average of 23.2 RE ppm across brands, and 18% were fortified below the standard minimum (<15 RE ppm) (Figure 2) with an average of 8.1 RE ppm across brands, 8, of which contained less than 10 RE ppm (5.8 ppm on average) (Figure 3A). Five of the analyzed brands were imported, of which one was not fortified, three were fortified above, and one was fortified below the standard minimum. Fifty-seven of the packaged brands (90%) were labeled with a fortification logo or statement, including 21 of the 26 brands that were not fortified.

The maximum peroxide value measured in the 66 packaged brands was 2.5 meg oxygen/kg, which was lower than the standard maximum (≤5 meg oxygen/kg soybean oil and ≤8 meg oxygen/kg palm oil) [23,24].

Of the 41 composite samples of bulk oil that were analyzed by bulk oil type, division and when possible by production sites and/or supplier, i.e., source (20 soybean, 13 palm oil and 8 super palm oil composites), 45% of soybean 31% of bulk palm oil, and 13% of super palm oil samples were fortified well below the standard minimum (Figure 2) with an average of 4.7, 3.6, and 3.5 ppm, respectively. One composite sample of soybean oil and two composites of bulk super palm oil were fortified according to standard (Figure 2) with an average of 19.5 and 18.4 ppm, respectively (Figure 3B).

Similarly, the maximum peroxide value measured in the 3 bulk oil types was 2.5 meg oxygen/kg, which is below the standard maximum (≤5 meg oxygen/kg soybean oil and ≤8 meg oxygen/kg palm oil) [22,23,25].

All the 33 packaged and labeled oil brands that were available in more than two divisions) were analyzed and confirmed fortified, and of these, 25 (76%) were confirmed to be fortified above the minimum of the fortification standard. Of the 52 packaged brands that were available in only one division, 33 were analyzed. While 27 of them were labeled as fortified, only 7 (21%) were confirmed fortified, and only 3 of these (9%) were fortified above the minimum of the fortification standard. Most industries supplying the brands that have lower availability across divisions adopted the fortification logo on their oil packaging; however, the fortification of most of these brands could not be confirmed.

The variety of brands is higher in the more populated divisions (such as Dhaka, Mymensingh, Chittagong, Rangpur and Rajshahi). While they make up a large number, most of these brands are small with a low spread and will, therefore, not make up a large proportion after adjusting by the supply volume. The less populated divisions have a lower number of available brands, and, therefore, larger brands that have a wide geographic spread make up a larger proportion of the available brands, and these have higher fortification quality (Figure 4A).

Fortification quality of bulk oil is poor across divisions and oil types (Figure 4D), with a few exceptions in some divisions where an adequate fortification content was measured in some bulk soybean and super palm oil (Figure 4B,D).

### 3.3. Market Volume

Approximately one-third of the total edible oil market volume (34%) in Bangladesh, based on production volume estimates, is sold as packaged oil brands (Figure 5). The 66 packaged brands analyzed represented about 96% of this volume (33% of the total market volume). The 40 fortified brands made up about 95% of the packaged oil volume: More than two-thirds (69% or 28 brands) of the packaged oil volume was fortified above the fortification standard minimum, and about one-third (28% or 12 brands) was fortified below the fortification standard minimum. The 26 packaged brands that were not fortified corresponded to about 5% of the packaged oil volume (2% of the oil market volume).

An estimated 66% of the oil market volume is sold in bulk. Fifty-nine percent of the bulk oil volume was not, and 41% was fortified, of, which about 7% was fortified according to standard.

The fortification quality of the assessed market volumes for both packaged and bulk oil (corresponding to about 99% of the total market volume) showed that over half of oil volume (59%) available in Bangladesh was fortified, close to a third (29%) above the standard minimum, and 41% of the oil market volume was not fortified.

## 4. Discussion

In this paper, we present the methods, data and interpretation of a market assessment of a mandated fortification food vehicle, i.e., edible oil, in Bangladesh and its fortification quality. The results of an assessment of this nature provide regulatory authorities and managers of the oil fortification program with an overview of the variety of packaged oil brands and bulk oil types available to populations living in different divisions of the country. Additionally, it facilitates the identification of the origin (producers) and supply channels of oil brands and quantifies the average vitamin A content of packaged oil brands and bulk oil types available at the market level and their compliance with the oil standards. Such information is critical for assessing program performance but also to guide further decisions in relation to potential equity issues.

With over two-thirds of the assessed oil volume being fortified, the prima facie conclusion is that the oil fortification program in Bangladesh is being implemented but not achieving its full potential for impact. This finding suggests smaller fortification and quality gaps (i.e., higher fortification quality overall) than other estimates documented in the literature [3,29]. However, we also observed a substantial difference in the fortification quality of packaged oil brands and bulk oils. While nearly all the packaged oil volume (95%) is fortified, less than half of the bulk oil volume is fortified. As bulk oil makes up a large proportion of the oil market volume, this means most of the oil volume available to the population is either not (25%) or fortified below the standard requirement (39%). Several conclusions and implications arise.

First, these findings permit the characterization of the edible oil market, identifying sources of heterogeneity and major bottlenecks in the fortified oil supply. Such information can inform the targeting or prioritization of inspections at production or import sites or further investigation of the supply chain (retailers, packers/repackers, warehouses, etc.). Clearly, the focus of fortification program managers and regulatory inspectors needs to be placed on the improvement of the fortification of bulk oil. How this (incentivizing businesses to fortify) can be best done is an area of further inquiry. To further elucidate the issues, we subsequently undertook value chain analyses of the reasons why companies do or do not fortify [30].

Second, assessing oil brand availability in several agglomerations with different population sizes (cities, towns, villages), as done here, gives an overview of the geographic spread of different brands and the kind of oil products available to different populations. Consequently, we can then identify disparities in brand variety in larger market hubs versus smaller and more rural places. In Bangladesh, most of the available oil brands are locally produced soybean, palm and rice bran oil, and we identified the leading eight brands with the widest geographic spread across urban and rural market hubs and divisions in the country. Additionally, the assessment provided an insight into the oil characteristics (brands, oil types, price category, packaging) and the oil product range available to people living and purchasing from retail outlets in the different market hubs. The product range of a retail outlet type can suggest oil availability to population groups purchasing from these types of retail outlets. These data can be used to match data on food vehicle product quality with data from the household level on food vehicle brand and type preferences and consumption patterns.

Third, the information collected from oil brand labels revealed that the majority of producers and suppliers were aware of the fortification program as they already adapted their packaging according to the labeling requirements. This level of awareness is a positive finding [31]. However, awareness and labeling do not always correspond with compliance with fortification requirements, as seen in these data.

Fourth, we expect that the three types of oil sold in bulk are primarily supplied by the same producers that also produce packaged and branded oils. Producers, packers, transporters, traders and retailers are expected to produce, handle and store packaged oil products appropriately to ensure the oil maintains its quality throughout the supply chain until it reaches the consumer. However, most of the composite samples of the bulk oil types by division showed no fortification or only very low vitamin A content. Bulk oil is usually transported and stored in third-party owned, recycled, non-food grade barrels. The non-food-grade packaging already raises food safety concerns. Furthermore, with bulk oil, one could plausibly assume that this type of handling may accelerate the oxidation process and accelerate degradation, thereby affecting the vitamin A content. However, the peroxide results did not indicate increased oxidation and, therefore, do not support this assumption [32,33]. The fortification results clearly suggest that a few local industries are fortifying the bulk oil in the same way as their packaged oil brands, which may be related to the modalities of the sale these industries apply to supply their brands to certain areas (supply in bulk rather than small packaging). The majority, however, seem to reduce or turn off the premix feeder for bulk oil production. As barrels are not labeled appropriately, the suppliers of the bulk oil cannot be directly traced back. The packaging and handling of bulk oil will need to be improved to meet oil quality standards and to ensure that the oil quality, including but not limited to the vitamin A content of bulk oil, is maintained until it reaches the consumer.

A key recommendation from this study is that oil producers and refineries need to be made accountable for the fortification of bulk oil by ensuring quality-conserving packaging and labeling that makes the oil traceable to their production site. Mandatory fortification law also applies to bulk oil; therefore, inspections of the fortification of producers/suppliers of bulk oil need to be prioritized and appropriate packaging and labeling reinforced by the regulatory authority to ensure traceability and quality that meets the oil standards. Through inspections of bulk oil processing at the production site, any causes and required improvements at the production level can be identified, and the importance of fortification of bulk oil emphasized. Additionally, traceability of unlabeled bulk oil specifically needs to be addressed, as the tracing of products identified as non-compliant back to the source is crucial for regulatory enforcement and enforcement of corrective action [34,35,36,37].

The assessment confirmed that bulk oils are the cheapest oils on the market and can be purchased in small quantities. It will, therefore, likely be the preferred choice for the poorer population groups, who have the highest risk of vitamin A deficiency [38]. To ensure that those most in need of additional vitamin A intake can benefit from the program, fortification program managers, regulatory inspectors and industry’s particular need to focus on the improvement of the fortification of bulk oil.

This market assessment produced program-relevant data at retail outlets, which are not only at a convenient handover node of the food vehicle to the consumer but also positioned in the middle of the impact pathway of the fortification program [2]. Results can, therefore, be used to inform the program managers in two directions: (a) identify gaps in the fortification of the food vehicle supply that require strengthening, which can be at the production or the supply chain level, and that can direct further investigation or inspections of the supply chain and trigger corrective action to improve fortification of the food vehicle, and (b) by combining oil availability and fortification quality, show the geographic spread of fortified food vehicle products through the number and proportion of the fortified packaged brands and bulk oil types available to consumers in different geographic areas.

If available brands are weighted by the estimated supply volume, they represent (as was done in this study), the proportion of the fortified food vehicle volume that is available to a population can be established and serve as a proxy for fortified food vehicle coverage in a population. The more detailed and accurate the available food volume data is, the better the coverage estimates. Moreover, in combination with survey data on household or individual consumption and coverage of oil brands or other oil criteria, such as oil types, packaging types and sizes, and the price is chosen by population groups, the market assessment data on fortified oil availability can be used to estimate the additional micronutrient intake and potential reduction of their micronutrient gaps of the population groups [5].

The market fortification assessment provides a standardized approach to measure fortified food vehicle availability at the market level. Such evidence is a critical complement to household access and consumption surveys [6]. The market information can also provide interim information on fortification program performance when resources for the higher-cost household surveys are not available or when questions exist on whether sufficient fortified food is available to warrant a coverage survey. Furthermore, the sampling and lab analysis by food product generates the average content that can be compared to the standard [6,39] and triangulated with the results of the same products inspected at production or import sites. If the objective is to ascertain the average content of food vehicles and the sources (suppliers and producers) of fortified, and particularly inadequately or unfortified products; or if the penetration of fortified foods is unknown; or if household coverage was recently assessed, this methodology is a better fit for purpose.

The market assessment conducted in Bangladesh comprised two components: brand availability and brand compliance with standards. Both components can be implemented separately. Once brand availability has been assessed, and available brands and oil products are known, fortified oil availability can be updated by solely repeating the compliance component, which only requires a condensed budget to visit a small number of markets and acquire brand samples for lab analysis to verify if brand fortification quality has evolved. Subsequently, estimates of additional micronutrient intake at the population level can be updated if data on amounts consumed are already available.

## 5. Conclusions

This market assessment revealed that most of the packaged oil brands are fortified, while big fortification quality gaps were seen in bulk oil that makes up a large proportion of the oil market volume, thereby limiting the impact of the oil fortification program in Bangladesh. This market assessment fills an information gap for managers and stakeholders of the edible oil fortification program in Bangladesh by (a) identifying oil brands and bulk oil types that do not meet the fortification standards and should be follow-up through further investigation or inspections to initiate corrective action, and by (b) presenting the geographic spread of fortified oil to urban and rural areas and across divisions of the country. Future efforts should focus on refining the methods to estimate the market share of brands in order to appropriately and reproducibly weight fortification quality available to the target population. Additionally, research and programming efforts should focus on the capacity of regulators and fortification program managers at the country level to appropriately conduct fortification quality assessments and interpret fortification compliance as part of regular monitoring.

## Figures and Tables

**Figure 1 nutrients-13-00794-f001:**
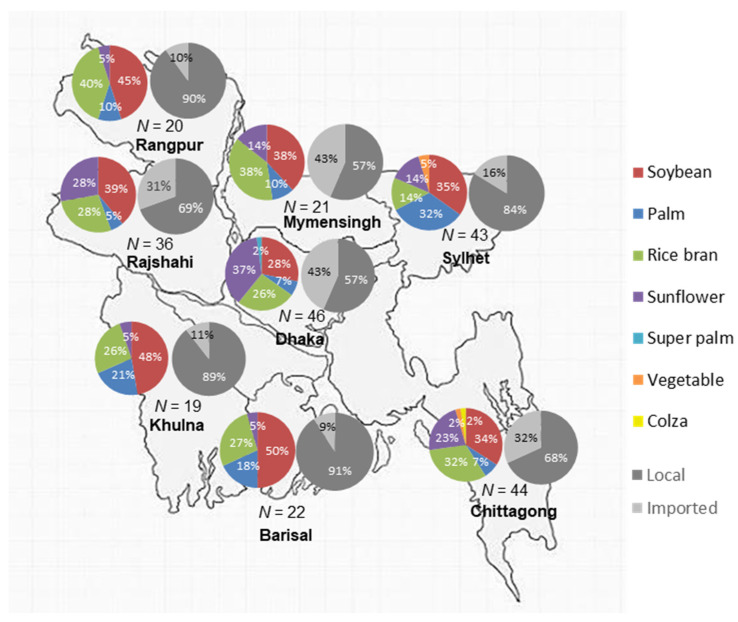
Proportion of oil types and origin of packaged oil brands available by division.

**Figure 2 nutrients-13-00794-f002:**
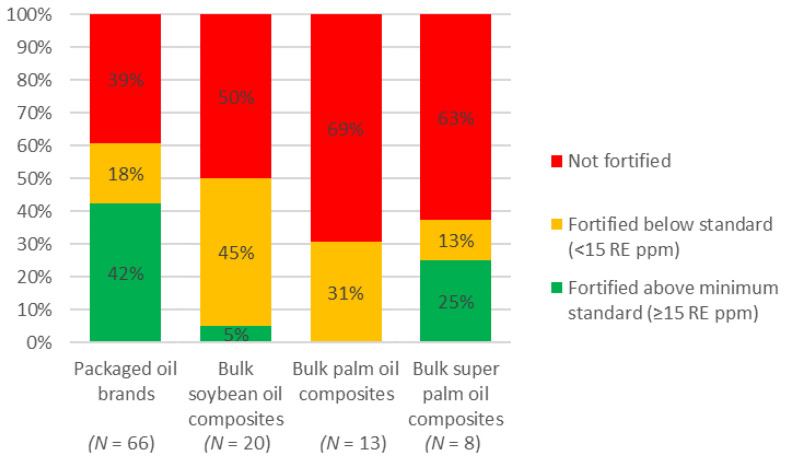
Proportion of fortification quality compared to Bangladesh national oil fortification standard (≥15 and <30 RE ppm) of packaged oil brands and bulk soybean, palm and super palm oil composites.

**Figure 3 nutrients-13-00794-f003:**
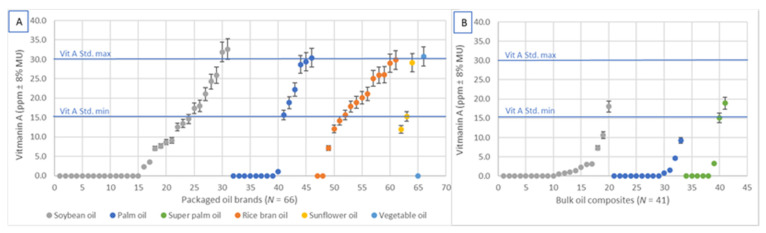
Average vitamin A content of packaged edible oil brands (**A**) and bulk oil composites (**B**) compared to Bangladesh national oil fortification standard (≥15 and <30 RE ppm).

**Figure 4 nutrients-13-00794-f004:**
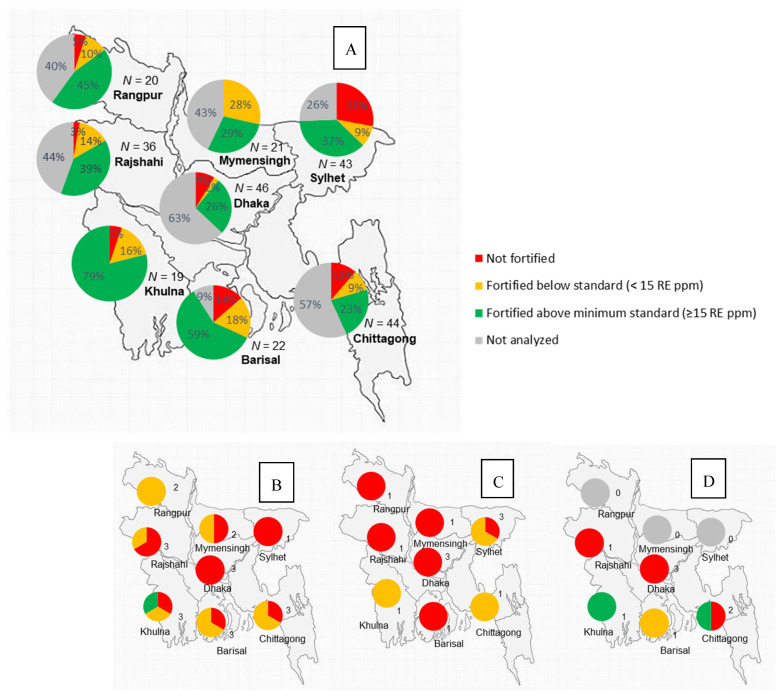
Number and proportion of fortification quality (vitamin A content compared to the Bangladesh national oil Figure 5. and <30 RE ppm)) of packaged oil brands (**A**) and bulk soybean (**B**) palm oil (**C**) and super palm oil (**D**) composites by geographical division.

**Figure 5 nutrients-13-00794-f005:**
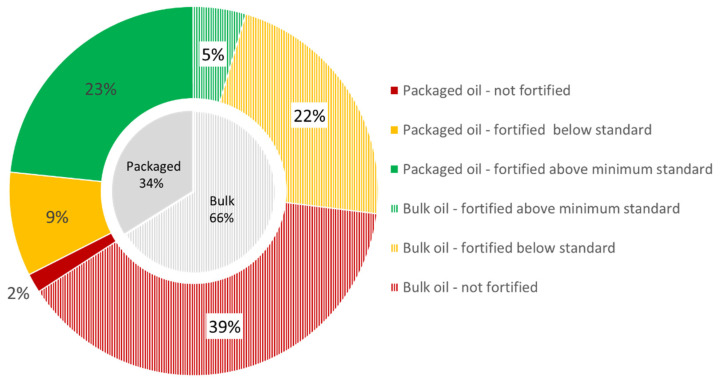
Estimated proportions of market volume of edible packaged and bulk oil in Bangladesh and fortification quality compared to Bangladesh national oil fortification standard (≥15 and <30 RE ppm).

## Data Availability

The data presented in this study are available on request.

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
