# Peer review of "Vitamin A Fortification Quality Is High for Packaged and Branded Edible Oil but Low for Oil Sold in Unbranded, Loose Form: Findings from a Market Assessment in Bangladesh"

_nutrients, 2021, doi:10.3390/nu13030794_

Round 1

Reviewer 1 Report

The paper is well written and it is about an important public health issue aimed to reinforce the need to monitor and correct the execution and performance of fortification programs, in this case, edible oil fortification with vitamin A. However, a major problem, as presented is the doubtful validity of the vitamin A analysis to determine the content of vitamin A in the samples. This needs to be fully clarified (page 4, specifically paragraph in lines 181-189. The extremely low vitamin A (RE) recovery in the samples of only 35% in comparison to the spike reference samples is striking. Why? What was going on? Furthermore, the determination and use of the 2.86 correction factor is not clear. This needs to be clearly explained and provide a reference to substantiate. The analyzed content of the spiked reference samples is not reported either and needs to be disclosed. There is also a need of describing the storage conditions and handling of the spiked reference samples to seek potential reasons for the poor recovery. Besides, is there any correlation between levels of peroxide values and vitamin content of the samples?. This is a critical issue in this paper that needs to be resolved.

Abstract. Data should be provided for the 3 categories of analyzed samples: a) Not fortified (n, %), b) Fortified below minimum vitamin A standard (n, %), and c) Fortified above minimum vitamin A standard (n,%). An important part of the conclusion should be that the majority of edible oil in the market is unfortified or below the required standard levels. 

Results. Imported oils were already fortified with vitamin A in the exporting country? How do the Vitamin A values compare to local oils? It is also important that for all vitamin A results indicate both the number of samples and the percentage (%) they represent in a way the reader can add up the figures to 100%. Regarding the graphs, the key indicators of colors often do not match all colors in the data, mainly because they cannot clearly be distinguished. This needs to be fixed.

Discussion. I do not believe, based on the data presented, that " The oil fortification program in Bangladesh is working reasonably well". I think it is all the contrary and this sentence needs to be deleted and be replaced for something else stating that the program in Bangladesh is not working properly and need to be fixed. The results of this study also need to be compared with those reported in other countries cited in the introduction (References 7-15). Is Bangladesh doing better or worst? Key performing factors in those countries if available? It also catches my attention reference 8 in the reference section which seems to be a similar or the same publication as the present article. What is or are the differences between this paper and reference 8? Is it a duplication?

Conclusion. Summarize results for the 3 sample categories (no fortified, below the minimum, and above the minimum). Emphasize that a large proportion of oils in Bangladesh is unfortified or fortified at levels below standard. And, needs to be fixed.

Author Response

Thank you for the constructive comments which we addressed. Please see the detailed responses in the attached document.

Reviewer 2 Report

Jungjohann et al. Present a good work about the quality of fortification of edible  oil with vitamin A in Bangladesh, comparing fortification  with Vitamin A packaged and  branded edible oil but vs oil sold in unbranded  and loose form.

Nevertheless the referee has a number of concerns that need an appropriate answer from the authors:

  1. Has the degree of purity of the oils been studied? In other words, has it been assessed whether there are traces of other products?
  2. Page 4, line 190: “Vitamin A degradation in oil is associated with lipid peroxidation. Peroxide value 190 was analyzed for the main oil brands and types of bulk oil to determine the degree of 191 oxidation that may be associated with low vitamin A content. The peroxide value was 192 measured in oil using a titration method (AOCS, 1998; Method No. Cd 8-53, AOAC 193 965.33).”. Why do you say that the peroxidation of the oils is associated with the low content of vitamin A? How can it be ruled out that the oxidation of the lipids is not due to impurities or to the manipulation of the samples in the places of origin apart from a low level of vitamin A content? Have the impurity levels been assessed? (authors may indicate purity levels in each oil sample).
  3. On figure 1 and 4, proportion in number must be introduced inside the figure, inside pie charts (similar to figure 5).

Author Response

Thank you for the constructive comments which we addressed. Please see the detailed responses in attached file.
